# Focusing on Autism Spectrum Disorder in Xia–Gibbs Syndrome: Description of a Female with High Functioning Autism and Literature Review

**DOI:** 10.3390/children8060450

**Published:** 2021-05-26

**Authors:** Stefania Della Vecchia, Roberta Milone, Romina Cagiano, Sara Calderoni, Elisa Santocchi, Rosa Pasquariello, Roberta Battini, Filippo Muratori

**Affiliations:** 1Department of Developmental Neuroscience, IRCCS Stella Maris Foundation, 56128 Calambrone, Italy; stefania.dellavecchia@fsm.unipi.it (S.D.V.); roberta.milone@fsm.unipi.it (R.M.); romina.cagiano@fsm.unipi.it (R.C.); sara.caldroni@fsm.unipi.it (S.C.); elisa.santocchi@fsm.unipi.it (E.S.); rosa.pasquariello@fsm.unipi.it (R.P.); filippo.muratori@fsm.unipi.it (F.M.); 2Department of Clinical and Experimental Medicine, University of Pisa, 56126 Pisa, Italy

**Keywords:** Xia–Gibbs syndrome, autism spectrum disorder, genetic autism

## Abstract

Background: Xia–Gibbs syndrome (XGS) is a rare disorder caused by de novo mutations in the AT-Hook DNA binding motif Containing 1 (*AHDC1*) gene, which is characterised by a wide spectrum of clinical manifestations, including global developmental delay, intellectual disability, structural abnormalities of the brain, global hypotonia, feeding problems, sleep difficulties and apnoea, facial dysmorphisms, and short stature. Methods: Here, we report on a girl patient who shows a peculiar cognitive and behavioural profile including high-functioning autism spectrum disorder (ASD) without intellectual disability and provide information on her developmental trajectory with the aim of expanding knowledge of the XGS clinical spectrum. On the basis of the current clinical case and the literature review, we also attempt to deepen understanding of behavioural and psychiatric manifestations associated with XGS. Results: In addition to the patient we described, a considerable rate of individuals with XGS display autistic symptoms or have been diagnosed with an autistic spectrum disorder. Moreover, the analysis of the few psychopathological profiles of patients with XGS described in the literature shows a frequent presence of aggressive and self-injurious behaviours that could be either an expression of autistic functioning or an additional symptom of the ASD evolution. A careful investigation of the abovementioned symptoms is therefore required, since they could represent a “red flag” for ASD.

## 1. Introduction

Xia–Gibbs syndrome (XGS) is a rare disorder discovered in the last decade caused by de novo mutations within a critical region of the AT-Hook DNA Binding Motif Containing 1 (*AHDC1*) gene, identified performing whole-exome sequencing. *AHDC1* function is still not fully understood: it may be involved in DNA repair as well as epigenetic and transcriptional regulation during axon genesis, with potential consequences on neurodevelopment [1].

This syndrome is characterized by global development delay, intellectual impairment, structural anomalies of the brain, global hypotonia, feeding issues, sleep difficulties and apnoea, facial dysmorphisms, and short stature [1,2].

Since its discovery in 2014 [3], approximately 250 cases of XGS have been reported until now, although the prevalence of this syndrome may be underestimated because of its phenotypic variability [1]. For this reason, the wide phenotypic spectrum of the syndrome has entailed molecular diagnosis as the primary diagnostic tool for XGS, instead of the recognition of distinctive facial dysmorphisms, or a facial gestalt. However, facial features are peculiar and include broad forehead, horizontal eyebrows, hypertelorism, down- or up-slanting palpebral fissures, depressed nasal bridge, thin upper lip, dysplastic dentition, and micrognathia [1,4].

Although the vast majority of mutations detected in *AHDC1* gene are truncating, specific genotype–phenotype correlations are difficult to identify, because of the XGS clinical heterogeneity. Indeed, patients may experience variable epileptic seizures, different brain abnormalities, various cognitive and linguistic profiles, and behavioural problems.

Here, we report an additional female patient whose phenotype deserves attention, since she does not present intellectual disability as most patients described in the literature, but shows a peculiar cognitive and behavioural profile indicative of high-functioning autism spectrum disorder (ASD). In addition, we describe her four-year follow-up in order to provide insight into her developmental trajectory and expand the XGS clinical spectrum. Finally, on the basis of the current case report and the review of the literature, we further explore a neglected issue, i.e., the behavioural and psychiatric pattern associated with XGS, with a particular focus on autistic features.

## 2. Methods

### 2.1. Case Report

Female patient of 7.5 years, followed since the age of 2 years and 4 months, diagnosed with Xia–Gibbs syndrome at the age of 6 years through Whole-Exome Sequencing (WES), studied through clinical observations and standardized scales aimed at defining a neuropsychological and behavioral profile. In this regard, as cross-sectional tools were used: the social communication questionnaire (SCQ) [5], the Autism Diagnostic Interview–Revised (ADI–R), a structured interview conducted with the parents [6] and the Autism Diagnostic Observation Schedule (ADOS)-2 Module 2, a standardised, semi-structured observational assessment of the child [7]. As longitudinal instruments, instead, were used: the Wechsler scales for assessing the intelligence (Wechsler Preschool and Primary Scale of Intelligence (WPPSI)-III; Wechsler Intelligence Scale for Children (WISC)-IV) [8], the Vineland Adaptive Behaviour Scales 2nd edition (VABS-II) addressed to a primary caregiver for the assessment of adaptive skills [9] and the Child Behaviour Checklist [10] to evaluate the psychopathological profile. We define spoken language based on the levels of expressive speech described in Tager-Flusberg and colleagues’ work [11]: Phase 1: Preverbal Communication, Phase 2: First Words, Phase 3: Word Combinations, Phase 4: Sentences, and Phase 5: Complex Language.

### 2.2. Literature Review

We conducted an electronic database search of Pubmed, Scopus and Web of Science until March 2021 using the following text word search strategy: *AHDC1* or Xia–Gibbs Syndrome. Two authors screened all titles and abstracts independently in order to exclude clearly irrelevant articles. Authors independently examined the full papers of the remaining articles to determine which articles met all inclusion criteria. The disagreements were resolved through mediation. After the database search, we examined the reference lists of the included articles for possible articles that were not located in the database search. We included in the review all articles reporting the clinical and/or neuroradiological phenotype of *AHDC1* variants, with no restrictions on the age or sex of the participants. However, we excluded articles that: (i) were duplicates; (ii) did not provide sufficient clinical information; (iii) were written in a language other than English or Italian; (iv) were conference abstracts or reviews; and (v) were clearly not related to our topic.

## 3. Results

### 3.1. Case Report: Clinical Data

The proposita is a 7.5 year-old girl (Table 1). She is the only child of non-consanguineous parents. Family history was negative for neuropsychiatric disorders. She was born at term after an uneventful pregnancy. Birth weight was 3650 g (50–75° centile), length was 49 cm (5–10° centile) and occipitofrontal circumference (OFC) was 36 cm (50–75° centile). Suction was valid, while swallowing and chewing were difficult. After 20 weeks of life, her height growth was consistently below the third percentile, while OFC maintained a constant growth curve.

She acquired sitting position at the age of 7 months, upright position at 16–17 months, and autonomous ambulation at 20 months. She presented delayed language development and produced only vowel sounds until the age of 30 months. She started increasingly to produce single words, association of words, and simple and complex sentences. Imitation was scarce and play schemes were prevalently sensorial.

Since the age of 6–7 months bilateral exotropia (more evident at left eye) with mild amblyopia and severe hyperopia had been detected. Strabismus had been surgically corrected at the age of 3 years.

Mild dysmorphic signs, including frontal bossing, low-set ears, preauricular tag, and smooth nasal philter, were noticed.

At the age of 2 years, she developed focal seizures with secondary generalization, initially during fever and later in apyrexia, successfully treated with carbamazepine. Electroencephalogram showed left focal paroxysmal activity and posterior slow activity on the right parieto–occipital regions. 

Brain MRI at the age of 28 months detected thin corpus callosum, incomplete left hippocampal inversion, and posterior fossa anomalies (retrovermian megacisterna magna, and left lateral and superior cerebellar cisterna dilatation). A second brain MRI at the age of 41 months confirmed these findings and in addition detected an asymmetric sulcation with prevalent infolding of precentral gyrus, without cortical dysplasia (Figure 1).

At the age of 4 years old, several examinations were performed for short stature (height <3 SD), with delayed bone age on wrist X-ray and deficiency of growth hormone (GH) and thyroxine. She started GH replacement treatment with clinical benefit. In fact, the auxological parameters at the age of 7.5 years show an improvement in statural growth with a height of 120 cm (25th centile), a weight of 26 kg (75th centile), and a head circumference of 54 cm (90–97th centile).

Electrocardiogram, echocardiogram, and abdominal ultrasound were normal. Auditory evaluation revealed mild bilateral conductive hearing loss.

Neurometabolic work-up was negative. Array-CGH analysis detected a 3p26.3 deletion (size 0.184 Mbp) inherited from her mother, not containing genes. FRAXA analysis resulted normal.

WES analysis showed a de novo truncating mutation in the *AHDC1* gene (c.514dupA, p.Ser172fs).

### 3.2. Case Report: Longitudinal Neuropsychiatric Profile

Neurocognitive and language aspects and behavioural manifestations are reported in Table 2. Neuropsychiatric profile between the ages of 3 and 4 years old was characterized by autism spectrum disorder, attention deficit, motor dyspraxia, and disharmonic intelligence profile, with better verbal performance. Reading both in Italian and in English language represented an adsorbent repetitive activity. She was continually distracted from written language and isolated from the context. She recognized numbers up to 9, enumerated up to 15, distinguished the main geometric forms and classified them according to color, shape, and dimension criteria. She did not accept being interrupted during the execution of her ritualistic activities. Eye contact was poor and difficult to be integrated with other communicative channels. Pointing was sporadic and it was usually used for requesting, not for sharing attention. Although she invested excessively in talking, her vocabulary was limited, her language was idiosyncratic, her prosody was peculiar, and her voice tone was loud. Language comprehension was below her chronological age. Echolalia and echopraxia were present. She did not pretend play. Motor and hand stereotypies emerged during that period, and head banging was evident when she was frustrated. Drooling was also present.

Verbal language has evolved positively over time. During the first evaluation at the age of 2 years and 4 months, verbal language was absent. At the age of 4 years and 2 months, a clear improvement in verbal language was detected, with the emergence of simple sentences [11]. At the age of 5 years and 7 months, the language assessment revealed a disharmonic language profile, with deficits in grammatical comprehension skills. Socio-communicative aspects were screened through the administration by the parents of the SCQ lifetime form, which refers to the individual’s entire developmental history. A total score of 16 was obtained from the completion of the questionnaire (cut-off score 15 [5]). ASD diagnosis was supported with the ADI-R, and the ADOS-2 performed at the age of 4 years and 2 months, showing a moderate level of symptom severity. Her general cognitive ability, as estimated by the WPPSI-III (band 2 years 6 months–3 years 11 months) at the age of 3 years and 6 months revealed a highly disharmonic intelligence profile, with average verbal abilities (Verbal Intellectual Quotient 120) and weak skills in performance (Performance Intellectual Quotient 52). This profile was confirmed by the administration of the comprehensive neuropsychological battery NEPSY-II [12] (Developmental Neuropsychology Assessment, Second Edition) at 4 years and 2 months, which showed visuo-graphic and visuo-motor difficulties in ‘Visuomotor Precision’, ‘Geometric Puzzles’, ‘Design Copying’, and ‘Visual Attention’ (this latter subtest was not completed by the child), against good skills in ‘Phonological Processing’, ‘Verbal Fluency’, and ‘Narrative Memory’. Cognitive level, evaluated with WPPSI-III scale [13] (band 4–7 years 3 months) at the age of 5 years and 7 months, was disharmonic, too, with average verbal abilities (Verbal Intellectual Quotient 94) and deficits of performance skills (Performance Intellectual Quotient 59) and processing speed (Processing Speed Index 52); visuo-graphic and visuo-motor abilities were significantly below average. At 7 years and 3 months, WISC-IV scale [8] showed average verbal abilities (Verbal Comprehension Index 94), average working memory (Working Memory Index 109), borderline performance skills (Perceptual Reasoning Index 71), and poor processing speed (Processing Speed Index 47).

The VABS-II administered by the parents was used to investigate adaptive skills. At the age of 4 years and 2 months, the VABS-II revealed average skills in communication (Intellectual Quotient, IQ, deviation: 103), borderline skills in socialization (IQ deviation: 75), and deficit in both motor skills (IQ deviation: 57) and daily living skills (IQ deviation: 64). At the age of 5 years and 7 months, adaptive skills were borderline in communication (IQ deviation: 78), socialization (IQ deviation: 70), and daily living skills (IQ deviation: 70) and poor in motor skills (IQ deviation: 51). Finally, at the age of 7 years and five months, adaptive skills were borderline in communication (IQ deviation: 74)—possibly due to the increased socio-communicative demands with increasing age—and poor both in socialization (IQ deviation: 67) and daily living skills (IQ deviation: 62).

Emotional and behavioral symptoms were evaluated through parents’ completion of the Child Behaviour Checklist (CBCL) [10]. At the age of 4 years and 2 months (CBCL 1 1/2–5), no scores in the clinical range were identified. Conversely, at the age of 7 years and 5 months (CBCL 6–18), borderline scores were obtained for internalizing problems (T = 61) and clinically significant score for externalizing (T = 72) and total problems (T = 71). As far as the Syndrome scales of the CBCL 6–18, clinically significant scores were obtained on ‘Thought problems’ (T = 71), ‘Attention problems’ (T = 68), and ‘Aggressive behaviour’ (T = 75), whereas clinically significant scores were obtained on ‘Affective problems’ (T = 73), ‘Attention deficit/hyperactivity problems’ (T = 78), and ‘Oppositional/defiant problems’ (T = 70) of the DSM-oriented scales. Thus, the longitudinal evaluation by means of the CBCL suggests that the behavioural profile worsened over time, with a prevalence of externalizing problems.

### 3.3. Literature Review

PRISMA Flow Diagram [14] of the review process is presented in Figure 2. The search of PubMed, Scopus, and Web of Science databases provided a total of 82 records. After removing duplicates, 38 articles remained for screening. After screening the abstracts, 19 articles were removed, including because they were conference abstracts (eight articles), literature reviews (one article), letters to editors (two articles), clearly not related to our topic (four articles), or were written in Chinese (one article). After full text screening, further articles were removed because they did not contain sufficient data about patient’s clinical phenotype (four articles).

Qualitative analysis of the 18 papers included in our review showed that a sufficiently detailed description of the clinical phenotype was available for 67 patients, information on neuroimaging normality or abnormality was available for 63 patients, and a detailed description of the neuroradiological phenotype was available for 31 patients. The clinical presentation of the 67 XG patients reported in the literature was heterogeneous, with high rates of neurodevelopmental problems/disorders (speech delay, motor delay, and intellectual disability), as well as hypotonia, facial dysmorphisms, and ataxia. In addition, other less frequent features were observed (see Figure 3). Autism was described in only 28.36% of cases. Analysis of the 63 patients, of whom at least one brain MRI was available, revealed altered structural neuroimaging in 70% of patients: thinning of the corpus callosum, posterior fossa abnormalities, and brain atrophy were the main neuroradiological findings reported (Figure 4).

## 4. Discussion

XGS is a highly heterogeneous syndrome, with a broad phenotypic spectrum of manifestations that may include epilepsy, brain anomalies with variable extensions, cognitive and linguistic profile widely distributed, and behavioural problems [15]. The complexity of both the clinical presentation and the gene in XGS implies that specific genotype-phenotype correlations are difficult to be established. Although less severe presentations have been hypothesized in association with heterozygous missense mutations in the *AHDC1* gene [4], emerging descriptions seem to disregard this assumption [3], including the current case report in which a mild clinical picture is in contrast with what would be expected in truncating mutations.

Our proposita has features typical of the XGS spectrum, such as global developmental delay, hypotonia with motor incoordination, short stature, facial dysmorphisms, mild hearing loss and visual impairments (strabismus and refractive defects), epilepsy, and brain anomalies. Different from the other XG patients already described, however, she does not present ID, and her cognitive profile resembles those reported in high-functioning (HF)-ASD. In particular, verbal versus non-verbal intelligence quotient (IQ) discrepancy rates have been frequently signaled in these individuals. Since cognitive intra-profile variability may invalidate or make not interpretable a global IQ score, it should be more meaningful to report discrepant scores instead of a misleading average score [16], especially in preschoolers [17], in order to underline weaknesses and strengths of patients, as we did. The Processing Speed Quotient, which reflects the child’s ability to work independently according to a given template, as well as mental flexibility/set shifting skills, sustained attention, and graphomotor speed/accuracy was lower compared to both verbal and performance IQ scores in our patient. Importantly, these skills are frequently impaired in ASD subjects and are considered predictors for executive and adaptive functions [18].

Besides the cognitive profile, also the linguistic profile of our patient resembles that of ASD subjects, with lower comprehension than production. Indeed, several studies observed not only that children with ASD often exhibit a discrepancy between language comprehension and production [19,20], but also that the prevalence of this pattern is more frequent for ASD individuals compared to both children with typical development and peers with other developmental disabilities [19,21].

The finding of VABS-II scores lower than the cognitive abilities may thus reflect executive functioning deficits typically associated with ASD, including planning and cognitive flexibility, even if difficulties in inhibitory control and sustained attention may also play a role [22]. Similar to other patients with high functioning ASD, adaptive behaviour skills appeared significantly lower than expected for cognitive abilities. In particular, the greatest weakness in socialization on the adaptive functioning profile is in line with what has been reported for ASD patients [23]. The worsening in social and communication adaptive abilities over time could be ascribed to the increase in social expectations with increasing age, when poor social awareness, deficits in the social aspects of language, difficulty in understanding non-verbal social signals, and difficulties in initiating and sustaining social interactions become more evident [24]. Indeed, according to the DSM-5 diagnostic criteria for ASD, ASD symptoms must be present in the early developmental period, but may not become fully manifest until social demands exceed limited capacities [25]. Although developmental abnormalities were evident in our patient before the age of 36 months, as revealed by the ADI-R [6] interview, autistic symptoms became more apparent over time.

Although data are still scant and inconclusive, the analysis of the literature revealed the relevant prevalence of neurodevelopmental disorders and psychiatric problems in individuals with XGS. Concerning neurodevelopmental disorders, a third of described cases are associated with ASD [3,26,27,28,29,30,31], in addition to motor/language delay and intellectual disability. In particular, approximately 29% of patients with sufficient information to derive a clinical phenotype description have been diagnosed with ASD [3,4,26,27,29,30,31]. Interestingly, previous studies [3,4] have also detected the presence of autistic symptoms, with a high risk of autism through the use of screening tools (M-CHAT score ranging from 8 to 20) in a relevant proportion of patients. Therefore, it would be necessary for these cases to investigate the potential ASD diagnosis using the appropriate clinical instruments (e.g., ADOS-2 and ADI-R).

By examining the Xia–Gibbs population, autism spectrum disorder or autistic traits prevalence was higher in males than females, with the same male to female ratio (4:1) reported in the literature of non-syndromic ASD [32]. Similar to what has been observed in other genetic conditions, males were more severely affected than girls concerning also other neurodevelopmental disorders, such as language impairment (i.e., the males are often nonverbal) [15] and ADHD. These data may further corroborate the hypothesis that males are genotypically less protected than females, and more vulnerable to neurodevelopmental disorders [33]. In addition, the comorbidity with ADHD—with prevailing attention deficit in our patient—was already reported in four patients with XGS (three males and one female) [2,26,27] in the literature review, and could be partly ascribed to common genetic basis between these two neurodevelopmental disorders [34]. Besides, ADHD may further exacerbate the impaired social functioning of ASD individuals. Our patient presented with epilepsy, too, as did another seven XG patients with ASD, confirming the frequent co-occurrence between these two disorders sharing etiopathogenetic factors, among which mutations in synaptic genes (i.e., synapthopathies) [35,36]. Furthermore, early-onset epilepsy is a risk factor for developing ASD itself [36], thus confirming a profound reciprocal interaction between these two disorders. In this framework, the severe pragmatic impairment detected in our proposita could reflect the strong association between early-onset epilepsy and ASD [37].

Various psychiatric problems have been described in patients with XGS, although there are no specific studies on this topic. The literature review we performed detected several behavioural problems, including aggressive behaviour or self-injurious behaviour [2,27,38,39,40], anxiety disorders [26,29,39], obsessive-compulsive disorder [26], impulse control difficulties [39], and schizophrenia during adolescence [40]. Aggressiveness, oppositional-defiant disorder (ODD) symptoms and mild self-injurious behaviours as reaction to frustration are also present in our patient. Such psychopathological features may be, however, an expression of the autistic functioning or could represent additional symptoms occurring in the evolution of ASD [41]. Therefore, the presence of the abovementioned symptoms deserves to be deepened by searching for ASD features not previously noticed [42]. Literature indeed highlighted that children with ASD have significantly more self-injurious behaviour [43], ODD symptoms [44], and aggressive behaviour [45] than peers with typical development

Regarding the motor aspects, the proband presented with a motor delay and subsequently clumsiness and motor incoordination, as most of the patients with XGS. Moreover, about half of patients with XGS described in the literature have been defined as ataxic. However, since ataxia is often a comprehensive term referring to incoordination, an accurate evaluation of this symptom over time should be performed in order to differentiate it from a developmental coordination disorder [46] or from the clumsiness frequently described in patients with ASD [47]. Of note, possible associations between motor incoordination and posterior cranial fossa abnormalities described in a non-negligible percentage of XG patients should be further explored, as they may reveal important brain–behaviour relationships.

Brain imaging has been poorly investigated and not well characterized so far in individuals with XGS. Based on the current systematic review of the literature, neuroimaging abnormalities are detected in the majority of patients, without preferential brain localization. The most frequent findings include thinning of the corpus callosum [2,4,15,26,27,30,48,49,50], posterior fossa anomalies [4,15,27,30,51], cerebral atrophy [26,31,39,40,49,51,52], white matter anomalies [2,30], and cortical malformation/dysmorphism [15]. Previous studies frequently observed a reduction of CC volume in subjects with ASD (see [53] for a review) as well as a correlation between decreased CC volume and severity of ASD features, supporting the hypothesis of impaired interhemispheric connectivity in ASD [54]. Crucially, the brain anomalies detected in our patient partially overlap with those previously described in literature. It could be important to further investigate the brain MRI data of XGS patients with the ultimate aim to better understand the complex interplay between genetic pathogenesis, brain dysfunctions, neuropsychological profile, and the behavioural clinical picture.

## Figures and Tables

**Figure 1 children-08-00450-f001:**
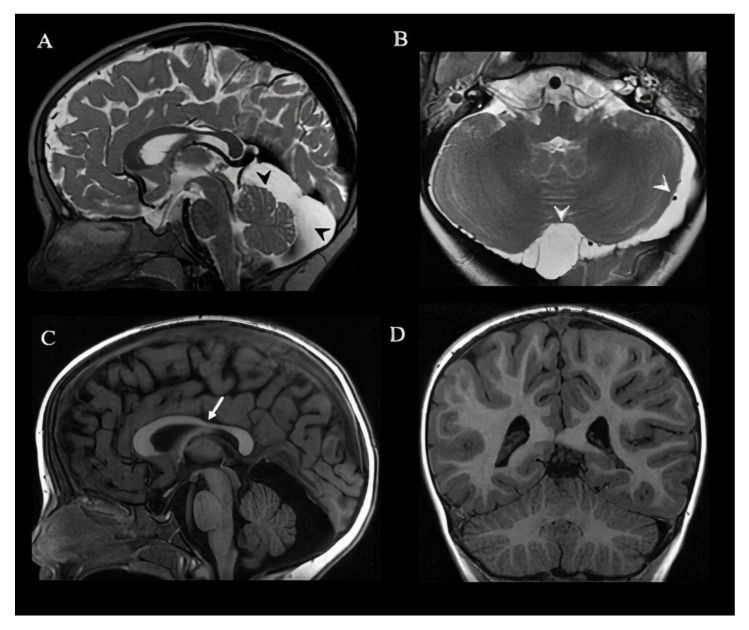
Brain MRI of a 3.5-year-old girl. Sagittal (**A**) and axial T2 weighted images (**B**) show enlarged retrovermian cistern and left pericerebellar subarachnoid space (black and white arrowheads). Sagittal T1 weighted image (**C**) shows thinning of the isthmus of the corpus callosum (white arrow). Sagittal (**A**,**C**) and axial (**D**) images show a normal conformation of the cerebellar hemispheres and vermis. MRI remained unchanged in comparison with the previous one performed at the age of 2.4 years old, except for a progression of the myelination.

**Figure 2 children-08-00450-f002:**
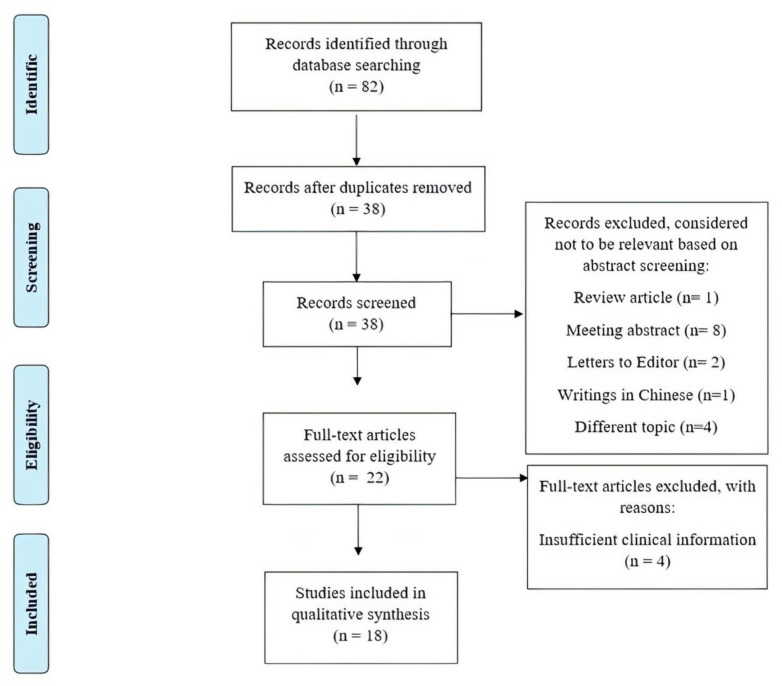
PRISMA flow diagram of search yield, screening and inclusion steps.

**Figure 3 children-08-00450-f003:**
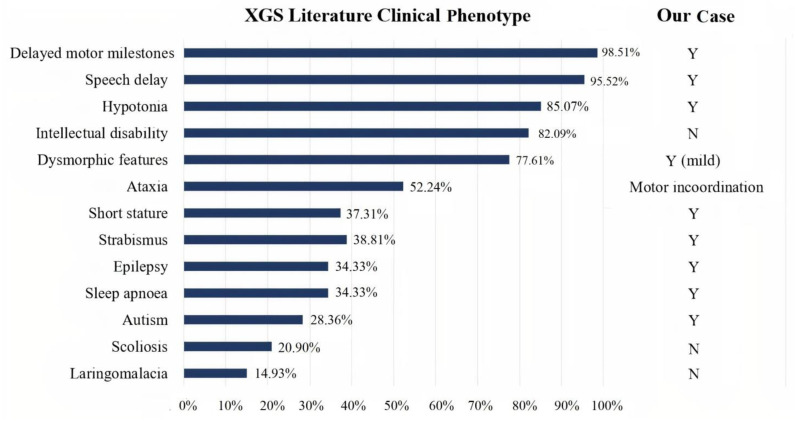
Bar graph showing XGS clinical phenotypes in 67 patients from the literature. Comparison between literature data (on the left), and our case (on the right). Y: yes. N: not present.

**Figure 4 children-08-00450-f004:**
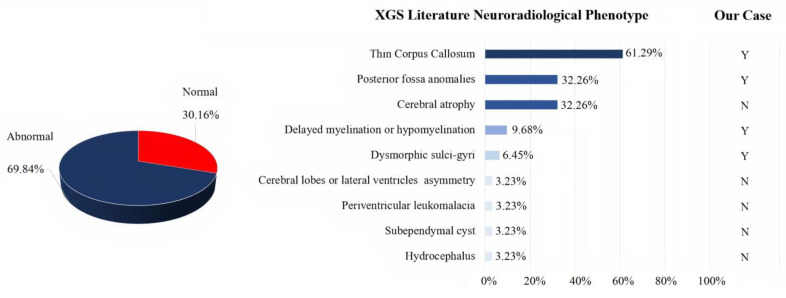
XGS neuroradiological findings in patient from the literature. Among a total of 63 patients with at least one brain MRI, 19 exhibited normal and 44 abnormal findings (pie chart). A detailed description of the literature brain MRI findings was only available for 31 of the 44 patients with abnormal findings in neuroimaging (bar graph). On the right-hand side, a comparison of our patient with literature data. Y: yes. N: not present.

**Table 1 children-08-00450-t001:** Genetic, clinical, and neuroradiological characteristics of our proposita.

**Genetic characteristics**		
Nucleotide and protein change	c.514dupA, p.Ser172fs	
Inheritance	de novo	
Age	7 yrs and 5 mths	
Gender	F	
Ethnicity	White	
**Neurological aspects**		
Hypotonia	Y	
Motor delay	Y (Independent walking at 20 mths)	
Language Delay	Y	
Motor incoordination	Y	
ID	N	
Epilepsy	Focal seizures with secondary generalization, successfully treated with carbamazepine	
Age of first seizure	2 yrs	
EEG	Left focal paroxysmal activity and posterior slow activity on the right parieto–occipital regions	
Brain MRI	TCC, posterior fossa anomalies and asymmetric sulcation without cortical dysplasia	
**ORL aspects**		
Sleep apnoea	Y	
Breathing support	N	
Hearing loss	Y (mild conductive)	
Adenotonsillar hypertrophy	Y	
**Vision aspects**		
Ocular Refractive Errors	Y (Hypermetropia)	
Strabismus	Y	
**Growth features**	At 4 yrs	At 7 yrs
HC	49 cm (50th Pc)	54 cm (90th Pc)
Weight	13.4 kg (<1.0 SD)	26 kg (75th Pc)
Height	87.8 cm (<3.0 SD)	120 cm (25th Pc)
GH deficiency	Y (GH therapy from the age of 4 yrs)	
**Orthopaedic aspects**	Absence of scoliosis and craniosynostosis	
**Dysmorphic features**	Low-set ears, flat nasal bridge, prominent frontal eminence	

yrs: years. mths: months. Y: yes. N: not present. TCC: thin corpus callosum. ORL: Otorhinolaryngology. HC: head circumference. SD: standard deviation. Pc: percentile. GH: Growth Hormone.

**Table 2 children-08-00450-t002:** Speech, Cognitive and Behavioural characteristics of the proband.

	3 yrs 6 mths–4 yrs 2 mths	5 yrs 7 mths	7 yrs 3 mths–7 yrs 5 mths
**Spoken language**	Sentences ***	Complex language *	Complex language *
**Cognitive skills**	**WPPSI-III** (band 2 yrs 6 mths–3 yrs 11 mths)	**WPPSI-III** (band 4–7 yrs 3 mths)	**WISC-IV** (band 6–16 yrs 11 mths)
	VIQ = 120 PIQ = 52	VIQ = 94 PIQ = 59 PSQ = 52	VCI = 94 PRI = 71 WMI = 109 PSI = 47
**VABS-II**			
Communication (IQ deviation)	103	78	74
Socialization (IQ deviation)	75	70	67
Daily living skills (IQ deviation)	64	70	62
Motor skills (IQ deviation)	57	51	-
**CBCL**	1 ½ -5yrs		6–18yrs
Internalizing (t-score)	60		61
Externalizing (t-score)	54		72
Total problems (t-score)	57		71
**SCQ-Lifetime**			
Total score	16 (cut-off 15)	-	-
**ADI-R**			
Social interaction score	12 (cut-off 10)	-	-
Communication score	13 (cut-off 8)	-	-
RRB score	7 (cut-off 3)	-	-
Abnormal development ^1^ score	>1 (cut-off 1)	-	-
**ADOS-2, Module 2**			
Social affect score	9	-	-
RRB score	4	-	-
Total score	13 (cut-off 7)	-	-

yrs: years. mths: months. * Spoken language was defined based on the levels of expressive speech described in Tager-Flusberg and colleagues. [11]: Phase 1: Preverbal Communication, Phase 2: First Words, Phase 3: Word Combinations, Phase 4: Sentences, and Phase 5: Complex Language. IQ: Intellectual Quotient; VIQ: Verbal Intellectual Quotient; PIQ: Performance Intellectual Quotient; and PSQ: Processing Speed Quotient. VCI: Verbal Comprehension Index; PRI: Perceptual Reasoning Index; WMI: Working Memory Index; and PSI: Processing Speed Index. RRB: Restricted and Repetitive Behaviours. ^1^ Abnormal Development evident at or before 36 months.

## Data Availability

Patient data are stored in the IRCCS Fondazione Stella Maris rare disease database. The data presented in this study are available on request from the corresponding author.

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
