# Peer review of "Focusing on Autism Spectrum Disorder in Xia–Gibbs Syndrome: Description of a Female with High Functioning Autism and Literature Review"

_children, 2021, doi:10.3390/children8060450_

Round 1
Reviewer 1 Report
It is a case study of a relatively rare disorder, supplemented by a literature review of that disorder. The manuscript serves the function of alerting professionals to this disorder and summarizing characteristics that have been reported. I particularly appreciated Figure 3, in which the authors summarized related symptoms. If this is to be a paper product, I'd delete the figure that describes the literature reduction efforts, but if the final product is all electronic, there is no reason to save space.
All in all, a nice job in describing a relatively rate disorder. I was unfamiliar with it, and I appreciate the work of the authors in affording me education.
Author Response
Thanks for your revision.
I modified our manuscript after an english revision
Reviewer 2 Report
This is a sound case report of an individual with a somewhat unusual presentation of Xia-Gibbs syndrome, accompanied by a sound literature review of reported cases of Xia-Gibbs syndrome that have a similar level of completeness with respect to description of the phenotype. An important aspect of the case report is that longitudinal data over the first 7 years of life is presented. The authors call attention to the somewhat atypical presentation of the syndrome in this patient, and suggest that the atypical characteristics might mandate further consideration of ASD features as concomitant issues in other individuals with genomically identified cases of Xia-Gibbs syndrome.
Another strength of this communication is the comparison of this case to the literature, as represented by a careful and well-defended literature review described in detail in this manuscript.
The authors are strongly suggested to obtain the assistance of colleagues fluid in the English language. The overall quality of the manuscript is reduced by multiple examples of poor English usage and even in some cases by missing or inappropriate words.
Author Response
Thanks for your suggestion.
We discussed and revised our manuscript with a Collegue expert in written english language.